# Mechano Chemical Compatibilization of Polyethylene with Graphite by Means of a Suitable Ester

**DOI:** 10.3390/polym15132770

**Published:** 2023-06-21

**Authors:** Simona Russo, Maria Rosaria Acocella, Annaluisa Mariconda, Valentina Volpe, Roberto Pantani, Pasquale Longo

**Affiliations:** 1Department of Chemistry and Biology, University of Salerno, Via Giovanni Paolo II, 132, 84084 Fisciano, Italy; sirusso@unisa.it (S.R.); plongo@unisa.it (P.L.); 2Department of Science, University of Basilicata, Viale dell’Ateneo Lucano 10, 85100 Potenza, Italy; 3Department of Industrial Engineering, University of Salerno, Via Giovanni Paolo II, 132, 84084 Fisciano, Italy; vavolpe@unisa.it (V.V.); rpantani@unisa.it (R.P.)

**Keywords:** mechanochemistry, compatibilization, polyethylene, carbon filler, non-covalent interactions, TGA analysis, X-ray diffraction

## Abstract

An effective non-covalent compatibilization method for graphite and low-density polyethylene is reported. To obtain this result, pyren-1-yl-stearate (P1S) was synthesized, characterized and mixed with graphite to provide a better dispersion in polyethylene matrix. The P1S improves the dispersion of carbon filler in polyethylene through non-covalent compatibilization: the pyrenyl group gives π−π stacking interactions with graphite and the stearyl chain provides van der Waals interaction with the polymer chain (specifically London dispersion forces). In this study, different P1S/graphite fillers were prepared with a ratio by weight of 90/10 and 50/50, respectively, by using manual and ball-milling mixing. Their stability, interaction and morphology were evaluated through TGA, RX, and SEM. Thermogravimetric analyses showed that ball-milling mixing is more effective than manual mixing in promoting π−π stacking interactions of molecules such as P1S ester containing an alkyl chain and aromatic rings. The role of ball milling is confirmed by X-ray diffraction measurements since it was possible to observe both exfoliation and intercalation phenomena when this technique was used to mix the P1S ester with graphite. SEM analyses of polyethylene containing 1% of the carbon fillers again highlighted the importance of ball milling to promote the interaction of the ester with graphite and, simultaneously, the importance of the alkyl chain in order to achieve polyethylene-graphite compatibilization.

## 1. Introduction

Polyolefins, a class of polymers that are easy to produce and low cost, are among the most used for their chemical–physical properties, such as low density, high stiffness and good tensile strength. They are eco-sustainable as they are non-toxic, biocompatible and easily recyclable [1,2,3,4]. However, for advanced applications, it may be necessary to further improve these properties by introducing new features. This can be achieved by adding nano-sized additives such as clays, silicas, carbon black, carbon nanofibers, graphene and carbon nanotubes [5]. Carbon additives were used not only to improve the mechanical properties, thermal stability and flame resistance of polyolefins but, above all, to give them thermal and electrical conductivity, a gas barrier, and electromagnetic and radiation shielding capabilities [6,7,8,9,10,11,12,13,14]. Among the carbon nanofillers, graphene appeared to be the most interesting because of its potential for a wide variety of applications due to its unique and extraordinary properties, i.e., flexibility, high thermal and electrical conductivity, large specific surface area and optical transparency. It is one of the allotropes of carbon and is the building block of all graphitic forms. Thus, graphene has the capability to be used in a wide variety of applications, albeit mainly in polymer nanocomposites [15]. Unfortunately, there are some limitations to these applications due to the lack of homogenous dispersion for the pronounced tendency to agglomerate because of its high surface energy. It has been shown that graphite can be directly exfoliated into monolayer sheets by dissolving it in certain solvents like *N*-methyl pyrrolidine (NMP), *N*,*N*-dimethylformamide (DMF) or o-dichlorobenzene (ODCB) [16], but this action is limited to solution mixing for nanocomposite preparation. Even in these conditions, graphene has a propensity to produce agglomerations. An alternative strategy to overcome this problem is surface graphite functionalization, which can ensure both easy exfoliation properties and better dispersion and interaction with the polymer matrix. The functionalization can be covalent or non-covalent. While the covalent functionalization usually destroys graphene’s conjugate structure compromising some of its properties, the non-covalent one allows the original graphene structure to be protected. Among the non-covalent modifications, such as van der Waals forces, electrostatic interactions, hydrogen bonds, and coordination bonds, the π−π stacking interactions are very attractive and easily realizable with planar aromatic graphitic structures. The ability of aromatic molecules, such as pentacene, [17], benzopyrene [18], 1-pyrenebutyrate [19], dopamine, melamine [20] and some other pyrene derivatives [16,21] to produce π−π stacking interactions with graphene sheets has been reported. The assembly of the polycyclic aromatic molecules with graphene was performed in water or an organic solution (ethanol, DMF, THF, etc.). More interestingly, the possibility of exfoliating graphite to provide stable graphene dispersion in many solvents by using an easy and eco-friendly ball-milling approach in the presence of melamine or triazine derivatives was previously reported by Leon et al. [20,22]. Through the application of mechanical forces, which are produced through pressing and shear forces of high-energy ball milling, aromatic organic molecules are first intercalated, thus reducing van der Waals forces between graphitic layers and stabilizing them using π−π stacking interactions, which in turn enables graphite to be exfoliated into graphene sheets. The rotational speed, number of balls, material and weight ratio between the balls and reagents all have profound effects on the efficiency of the procedure. Here, we show the ability of functionalized pyrene with a fairly long alkyl chain to be first intercalated into the graphitic planes and then provide partial graphite exfoliation to be used as a filler in PE nanocomposite via the melt-compounding process. In fact, the industrial applicability of this approach with the respect to solution mixing, in situ polymerization or melt blending from a solvent-casting masterbatch is well known because of its high efficiency, easy scale-up, and lack of solvent request during processing.

## 2. Materials and Methods

All the reagents were bought from Merck Darmstadt, Germany or TCI Europe Zwijndrecht Belgium. Stearoyl chloride (>97%, TCI), hydroxypyrene (>98%, TCI), pyridine dry (99.8%, Merck, Darmstadt, Germany), graphite (C 99.8 wt%, Asbury graphite Mills Inc. 156 Asbury-West Portal Road. Asbury, NJ, USA), polyethylene (low density, Merck, Darmstadt, Germany) and pentane were used as received, whereas dichloromethane was anhydrified before use.

All the reactions were conducted under a nitrogen atmosphere using a glovebox and Schlenk techniques. The products were characterized via ^1^H and ^13^C NMR spectroscopy. The samples were prepared by solubilizing about 10 mg of products in 0.5 mL of deuterated solvent (Eurisotop Cambridge Isotope Laboratories, Cambridge, UK).

For NMR analysis, a Bruker AM 300 (Milano, Italy) spectrometer (300 MHz for ^1^H; 75 MHz for ^13^C was used. ^1^H and ^13^C chemical shifts are given in parts per million (ppm), downfield from TMS, and are referenced from the solvent peaks or TMS. Chemical shift (ppm), multiplicity and integration were specified for each signal from the ^1^H and ^13^C NMR spectra. Multiplicities have been abbreviated as follows: singlet (s), doublet (d), triplet (t), multiplet (m) and broad (b).

Elemental analysis was conducted using a PERKIN-Elmer 240-C analyzer (Waltham, NJ, USA).

A planetary ball mill Pulverisette 7 Premium (Fritsch GmbH, Germany) at room temperature was used for the ball-milling experiments that were conducted using a silicon nitride jar in which eight silicon nitride balls with a diameter of 10 mm were introduced. The rotational speed of 300 rpm was set for one hour with breaks every 5 min.

TGA measurements were carried out using a Q500 TA Instruments, Milano, Italy). Samples of 5 mg were placed in platinum pans, and experiments were conducted in nitrogen, from 25 °C to 500 °C, at a rate of 10 °C min^−1^.

X-ray diffraction analyses of the powder samples were performed using an automatic Bruker D8 Advance diffractometer in reflection with the nickel-filtered Cu-Ka radiation (1.5418 Å).

Synthesis of pyren-1-yl-stearate: The synthesis of pyren-1-yl-stearate was carried out under a nitrogen atmosphere. A solution of 0.720 g of 1-hydroxypyrene (3.30 mmol) in 110 mL of dichloromethane was introduced in a 250-mL 2-necked flask. First dry pyridine (0.500 mL, 6.60 mmol) and then stearoyl chloride (0.997 g, 3.30 mmol), dissolved in a small amount of solvent, were added dropwise into the same flask. The reaction was stirred in an ice bath for 30 min and then at room temperature for 3.5 h.

After removing the reaction solvent, pentane was added to the reaction mixture and, pyridinium chloride, which precipitated as a white solid, was separated through filtration. The ester was recovered as a pale-yellow solid after the removal of the solvent under the vacuum. Yield: >90%.

^1^H NMR: (300 MHz, CD_2_Cl_2_, δ): 1.26 (s, 3H, (CH_2_)_12_CH_2_CH_2_C*H*_3_), 1.91 (s, 2H, (C*H*_2_)_12_C*H*_2_C*H*_2_CH_3_); 2.33 (m, 2H, C*H*_2_(CH_2_)_12_); 2.83 (m, 2H, OC=OC*H*_2_), 7.78, 8.09–8.22 (aromatics).

^13^C NMR: (75 MHz, CD_2_Cl_2_, δ): 14.3 ((CH_2_)_12_CH_2_CH_2_*C*H_3_), 23.1 ((CH_2_)_12_CH_2_*C*H_2_CH_3_), 25.4 (*C*H_2_(CH_2_)_12_), 30.16 ((*C*H_2_)_12_), 32.4 ((CH_2_)_12_*C*H_2_CH_2_CH_3_), 34.5 (OC=O*C*H_2_), 120.4–145.0 (aromatics), 173.1 (O*C*=OCH_2_).

^1^H and ^13^C NMR spectra are reported in the Appendix A, respectively as Appendix A.

Elemental analysis: calcd. for C_34_H_44_O_2_: C, 84.25; H, 9.15; O, 6.60. Found: C, 84.00; H, 9.20; O, 6.80.

Graphite/pyren-1-yl-stearate and graphite/1-hydroxypyrene (1H), both manually mixed and ball mixed, were added at 1% by weight to a low-density polyethylene using a micro compounder with conical, counter-rotating and intermeshing twin screws from Haake MiniLab II (Thermo Fisher Scientific, Schwerte, Germany), at a temperature of 120 °C, a screw rotation of 30 rpm and a cycle time of 5 min.

Morphological characterization was carried out by adopting a desktop SEM (Phenom ProX, Phenom-World BV, Eindhoven, The Netherlands). Before the analysis, the samples were coated with a thin film of gold via sputtering. The micrographs obtained from SEM observations were analyzed using a freeware for image analysis, ImageJ (1.54a, Kensington, MD, USA) [23]. At least two samples for each condition were analyzed.

## 3. Results

### 3.1. Synthesis of Pyren-1-yl-Stearate (P1S)

The synthesis of pyren-1-yl-stearate was carried out through a reaction of 1-hydroxypyrene with stearoyl chloride in a solution of dichloromethane and in the presence of pyridine. The ester P1S was recovered after the filtration of the pyridinium chloride followed by the evaporation of the solvent. The reaction scheme is shown in Figure 1.

### 3.2. Preparation of Graphite/Pyren-1-yl-Stearate (P1S) Samples

A total of 100 mg of several samples of graphite/pyrene-1-yl-stearate were prepared at two different weight percentages: 90–10, 50–50. For each chosen composition, two samples were realized through manual mixing (MM) and ball milling (BM). In addition, 50 wt% of 1H was mixed with 50 wt% of graphite through manual mixing (MM) and ball milling (BM). All the prepared samples were subjected to thermogravimetric analysis and X-ray diffraction spectroscopy and, subsequently, used as fillers for the realization of polyethylene matrix-based composites.

### 3.3. Preparation of Polyethylene-Based Composites

Polyethylene-based sample filaments, containing the filler at 1% by weight, were obtained through melt extrusion. In order to observe the samples through scanning electron microscopy (SEM), 250-μm-thick films were produced from the filaments through compression molding at a temperature of 120 °C and a compression time of 3 min. Below we list the fillers that we used to make the samples:manually mixed (MM) and ball-milled graphite (BM),manually mixed (MM) and ball-milled (BM) graphite/pyrene-1-yl-stearate (P1S) at 90–10, 50–50 weight percentages,manually mixed (MM) and ball-milled (BM) graphite/hydroxypyrene at 50–50 weight percentages.

## 4. Discussion

Incorporating carbon fillers into polyethylene-like systems is important in improving the mechanical and electrical properties of the systems, but it is challenging because of their different chemical nature.

Aiming to improve the compatibility of graphite with polyethylene and reduce the tendency of graphite particles to agglomerate into the polymer matrix, we synthesized a molecule containing a pyrene group and a hydrocarbon chain, the first capable of interacting with graphite and the latter, despite its short length (18 carbon atoms), capable of partially simulating a polyethylene chain.

Firstly, the most important interaction we intended to exploit was non-covalent. It was assumed that a π–π stacking interaction may be established between the aromatic rings of graphite and those of the pyrene group.

For this reason, the pyren-1-yl-stearate (P1S) was synthesized through a acyl nucleophilic substitution reaction between 1-hydroxypyrene and stearoyl chloride.

To prevent the transformation of acyl chloride into carboxylic acid, the synthesis was carried out under a nitrogen atmosphere.

An excess of pyridine was added to the 1H solution before the introduction of the stearoyl chloride to prevent the ester hydrolysis. As a result of the reaction between 1H and stearyl chloride, as soon as the formation of hydrochloric acid occurred, the acid was immediately neutralized by pyridine, already present in the reaction environment, which gave rise to pyridinium chloride. At the end of the reaction, the solvent was removed from under the vacuum and the product was washed with pentane; the desired product was soluble in this solvent, while the insoluble pyridinium chloride was removed through filtration.

The ester formation was confirmed through ^1^H NMR analysis because of the change of the resonances related to the methylene protons in the alpha position with respect to the carbonyl; specifically, the signals attributable to the methylene protons in stearoyl chloride are at 2.89 and 1.68 ppm, while they move to 2.83 and 2.33 ppm for the methylene protons of the pyren-1-yl-stearate.

Furthermore, in the ^13^C NMR spectrum, the shift of the diagnostic resonance of the carbonyl from 174.2 in the stearoyl chloride to 173.1 in the pyren-1-yl-stearate is further evidence that the reaction yielded the desired product.

The formation of the ester molecule was also proved through elemental analysis.

The greatest challenge was to have the ester molecules enter between the graphitic layers and facilitate the establishment of innumerable π–π stacking interactions between the graphite and pyrenic groups; on the other hand, the alkyl chain can be more compatible with polyethylene.

As previously reported for benzopyrene [18], it is possible to assume that a molecule as small as hydroxypyrene can easily enter between the graphitic layers. Conversely, if an alkyl chain is bonded to a pyrene group, such as the P1S ester, this may prevent the molecule from intercalating.

Hence, to evaluate the ability of P1S to be intercalated in graphitic layers, a commercially available graphite was mixed with the synthesized pyrene-1-yl-stearate P1S at different weight percentages: 90–10% and 50–50%. Additionally, a 1H/graphite mixture at 50–50% by weight was prepared to compare intercalation abilities.

In order to promote the intercalation of polycyclic aromatic derivatives and verify the effectiveness of the ball-milling technique for this purpose, for each chosen composition, the samples were realized through manual mixing (MM) and ball milling (BM).

The main role of ball milling is to reduce the particle size thanks to the collision of the spheres, producing a homogeneous mixture of different materials and leading to the establishment of interactions between molecules. For this reason, it was thought ball milling could play an important role for our purposes.

It is important to note that results of the ball-milling technique can be optimized by adjusting the milling time or speed and the ball size. In this perspective, we established that the best conditions were those described in the Material and Methods, Section 2.

Thus, a comparison, through thermogravimetric analyses and X-ray diffraction measurements, of the samples obtained via ball milling and manual mixing was central to this work.

### 4.1. Thermogravimetric Analyses

In the picture below, the overlay of the thermograms of the samples obtained by both MM and by BM of graphite/pyren-1-yl-stearate both in the ratio 90/10 and 50/50 and the samples obtained by MM and by BM technique of graphite/1H in the ratio 50/50, are reported. Moreover, for a convenient comparison, the thermograms of graphite, 1H, and pyren-1-yl-stearate are also reported (Figure 2).

1H completely degrades at 241.9 °C, whereas the P1S ester degrades at 355.7 °C. This indicates that the presence of the hydrocarbon chain, bonded to the aromatic rings in the P1S ester, increases its thermal stability. On the other hand, graphite maintains its structure and properties at temperatures up to 500 °C.

TGA scans of manually processed and mixed samples revealed interesting results.

The thermogravimetric curves of samples containing 90 wt% of graphite and 10 wt% of the P1S ester showed an 8% loss in weight for the milled samples, while the loss was 10% for the unmilled one. The effect was even more pronounced in the presence of samples containing 50 wt% graphite and 50 wt% ester; in the case of the milled sample, a 40% (41.7) weight loss occurred, while in the case of the unmilled sample, the loss was 50% (o 49.8). These data points are summarized in Table 1. The temperatures are referred to the maximum degradation rate assessed at the inflection point, while the weight losses (%) to the final degradation.

Based on these results, we can infer that in the case of the graphite-P1S ester samples, there was a percentage of the ester that was protected when the samples were milled, while such did not occur when they were prepared through manual mixing. Additionally, the behaviour of the samples obtained by mixing 50 wt% of graphite with 50 wt% of 1H through both manual mixing and ball milling was studied. Even in this case, the trend was respected because the curve of the milled sample was above that of the manually mixed one.

From the thermogravimetric analyses, it is possible to deduce that the P1S ester is capable of intercalating the graphene layers only if the ball-milling technique is used.

In the case of the graphite/P1S ester sample at 50–50% by weight, prepared through the ball-milling technique, more than 10% of the ester remained undegraded up to over 500 °C, while it was completely degraded at the same temperature as the free ester (about 370 °C) when manually mixed. Small molecules, such as hydroxypyrene, can intercalate graphite via mixing both through the ball-milling technique and manually. In both samples the almost complete degradation of the hydroxypyrene was evident at temperatures around 270 °C. Thermogravimetric analyses allows us to state that the hydrocarbon chain prevents the polycyclic aromatics from escaping from the graphite structure when it has intercalated the graphene planes as a result of the ball-milling technique.

### 4.2. X-ray Diffraction Measurements

The samples have also been analyzed through X-ray diffraction spectroscopy (Figure 3).

In the figure above, the profile of the pristine graphite and the graphite/P1S samples at different compositions is shown. The X-ray diffraction pattern of graphite highlights the characteristic 002 reflection due to the interlayer distance between the graphitic planes (d_002_ = 0.34 nm) and the 100 and 101 reflections corresponding to the in-plane periodicities. By comparing the profiles of graphite/P1S ester at different compositions, a significant amorphization clearly appears in the samples subjected to ball milling, as the percentage of the ester is increased up to 50% of P1S ester in the sample. The 100 and 101 reflections related to the in-plane order show the same intensity for all the samples reported in Figure 2, with a slight reduction just in the 50/50 graphite/P1S ester compounds, manually and ball milled, (inset B Figure 2) due to the progressive increase of disorder [24]. Additionally, the 002 reflection shows a reduction of half-height width and a shift to lower and higher theta for manually mixed and ball-milled mixed samples, respectively (inset A, Figure 2). Moreover, these effects, already present for graphite, become more pronounced for graphite/P1S ester 90/10. This is a new and very interesting effect compared to what typically happens to crystalline graphite after ball milling. Specifically, it has been reported that the crystalline graphite usually exhibits a reduction in particle size and graphene layer count as a consequence of dry grinding treatment [25,26]. On the other hand, our results suggest that milling graphite with a high surface area (currently used in our studies) increases the amorphous halo and simultaneously enables the aggregation of graphene layers, thereby reducing the half-height width and shifting the 002 reflection to a higher theta. As the graphite/P1S ratio increases up to 50/50, these phenomena disappear due to the intercalation of ester in the graphitic structure. Additionally, the increasing of the amorphous halo in the 50/50 graphite/P1S compounds can be explained by the ball milling action, which, assisting the ester to be inserted into graphitic planes, induces exfoliation and destroys the order. It is worth noting that no amorphization was observed for the ester after 1 h of grinding as its crystalline structure was unchanged, as seen from its diffraction pattern superimposed on its profile (Figure 2). A double effect of exfoliation and intercalation is, therefore, considered evidence of the interaction between ester and graphite.

### 4.3. Scanning Electron Microscopy Analyses

Multiple polyethylene films carrying 1% of the filler consisting of 50–50 wt% of the aromatic additive and graphite were prepared. These were previously generated through either manual mixing or milling. Scanning electron microscopy (SEM) was performed to verify the successful incorporation of the fillers into the polymer matrix.

As reported in the SEM images (Figure 4b,d,f vs. Figure 4a,c,e, respectively), it is clear that the ball milled samples show a better dispersion of the fillers, regardless of whether graphite alone, graphite/P1S, or graphite/1H is present.

To better evaluate the effect of the alkyl chain on the possible compatibilization of the ester on the filler dispersion, a software for image analysis was adopted to analyze the particle size distribution in SEM micrographs. Following the application of a series of filters, the thresholding or binarization method was adopted to measure the areas occupied by the particles. Since graphite particles have a lamellar shape, Feret’s diameter was adopted as a characteristic size. Feret’s diameter is commonly used in the analysis of particle sizes for projections of a three-dimensional object on a 2D plane and is defined as the distance between the two parallel planes restricting the object perpendicular to that direction. The particle size distribution was obtained by evaluating the percentage of area occupied by particles of a certain Feret’s diameter in relation to the total area occupied by the particles. The Feret’s diameters were divided in 10 ranges of 0.1 micron, up to a maximum diameter of 1 micron. In order to evaluate the effect of ball milling, the difference between the particle size distribution in PE with graphite/P1S 50/50 manually mixed and ball mixed was measured.

As a result, the particle size distribution of PE-graphite, PE-graphite/P1S 50/50 and PE-graphite/1H 50/50 BM (Figure 5) clearly shows the better dispersion of the PE-graphite/P1S 50/50 BM, since the percentage of particle size in the range of 0–0.2 micron is higher. Therefore, it appears that the hydrocarbon chain plays a critical role in the compatibility of fillers with a polymer matrix.

Additionally, a further comparison of the particle size distribution of PE-graphite/P1S 50/50 MM and BM was performed (Figure 6), showing once again the positive effect of the milling treatment on the filler’s dispersibility.

## 5. Conclusions

In this work, a pyren-1-yl-stearate (P1S) was synthesized with the aim to improve the compatibility of graphite with a low-density polyethylene by reducing the problem of the agglomeration of the graphite particles. The π−π stacking interaction between pyrenyl groups with planar aromatic graphitic structures and van der Waals interactions between polyethylene and stearyl chain made the system much more homogeneous. The pyrene group had a strong interaction with graphite because it could enter between the graphitic layers. This occurred when the mixing between the polymeric matrix, the compatibilizer and the graphite filler was carried out using the ball-milling technique.

Thermogravimetric analyses made it clear that when the graphite/P1S samples were prepared through the ball-milling technique, there was an amount of ester which was protected, probably due to the intercalation of the pyrenyl groups in the graphitic layers. This did not occur when samples were obtained through manual mixing. This suggests the importance of the role of the ball-milling technique to obtain the intercalation of the P1S ester in the graphite layers and therefore the best compatibilization.

An additional proof of interaction of P1S ester with graphite layers was obtained through X-ray diffraction analyses that clearly showed exfoliation and intercalation following milling treatment.

The SEM images of both milled and manually mixed compatibilizer samples revealed that ball milling was crucial for the intercalation and better dispersion of the compatibilizer, resulting in a more homogeneous material with a graphitic filler in polyethylene. Our preliminary studies show the positive effect of the ball-milling technique to improve the filler dispersion of polyethylene/graphite composite.

## Figures and Tables

**Figure 1 polymers-15-02770-f001:**
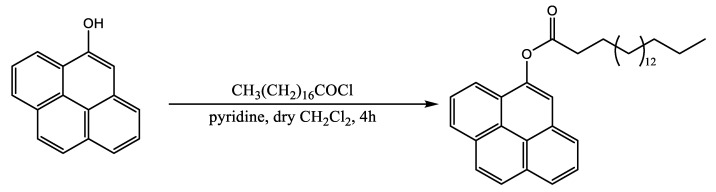
Synthesis of pyren-1-yl-stearate (P1S) ester.

**Figure 2 polymers-15-02770-f002:**
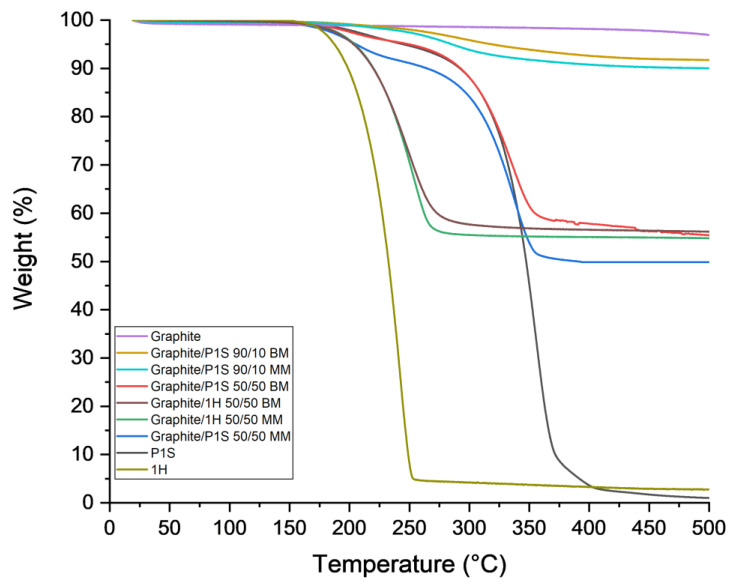
Thermogravimetric curves of the milled and not milled samples.

**Figure 3 polymers-15-02770-f003:**
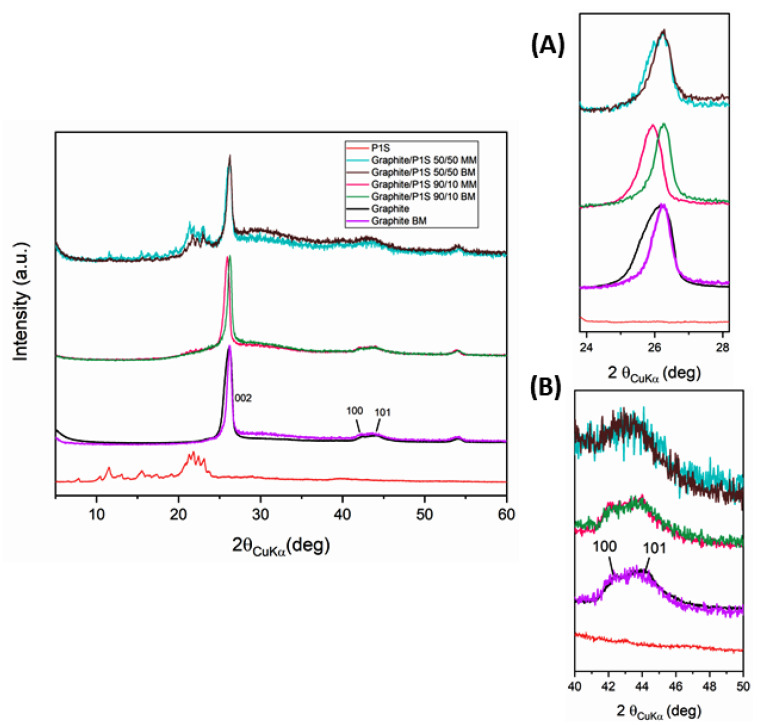
X-ray diffraction pattern of P1S ester (red), starting graphite (black), milled graphite (violet) and graphite/P1S ester at different compositions: 90/10 manually mixed (fuchsia) and milled (green), 50/50 manually mixed (light green) and milled (black). Insets (**A**) and (**B**) show enlarged patterns for 2θ ranges, 24–28 and 40–50, respectively.

**Figure 4 polymers-15-02770-f004:**
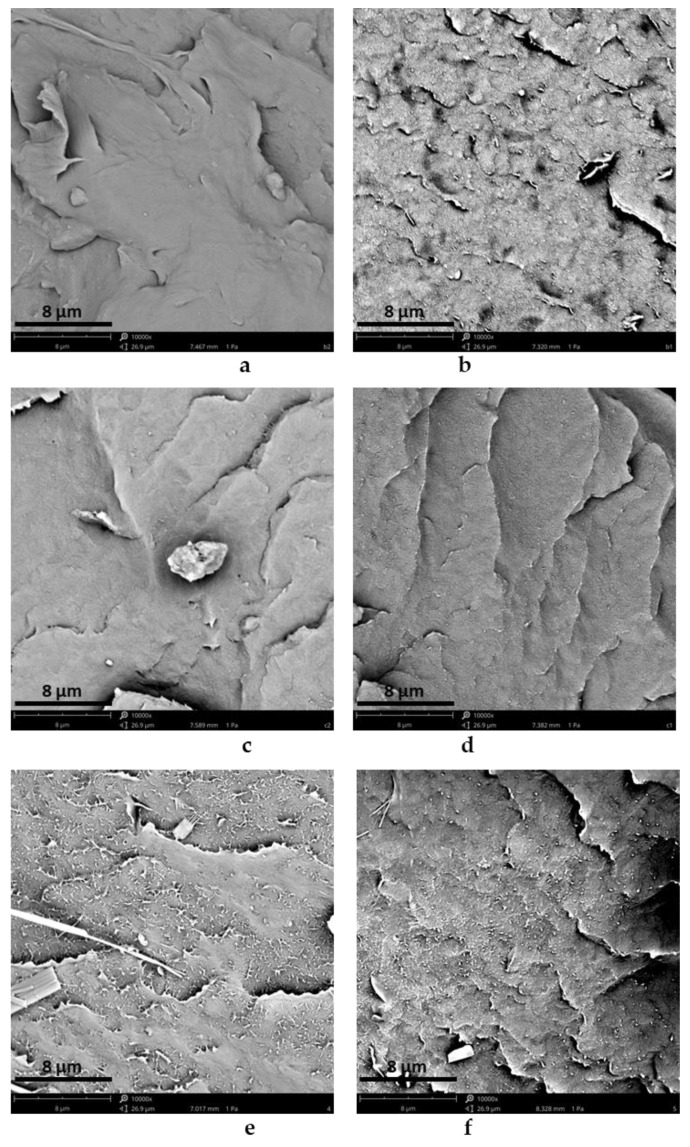
Scanning electron microscopy of (**a**) PE-graphite MM, (**b**) PE-graphite BM, (**c**) PE-graphite/P1S 50/50 MM, (**d**) PE-graphite/P1S 50/50 BM, (**e**) PE-graphite/1H 50/50 MM, (**f**) PE-graphite/1H 50/50 BM. Image dimensions: 26.95 × 28.63 μm.

**Figure 5 polymers-15-02770-f005:**
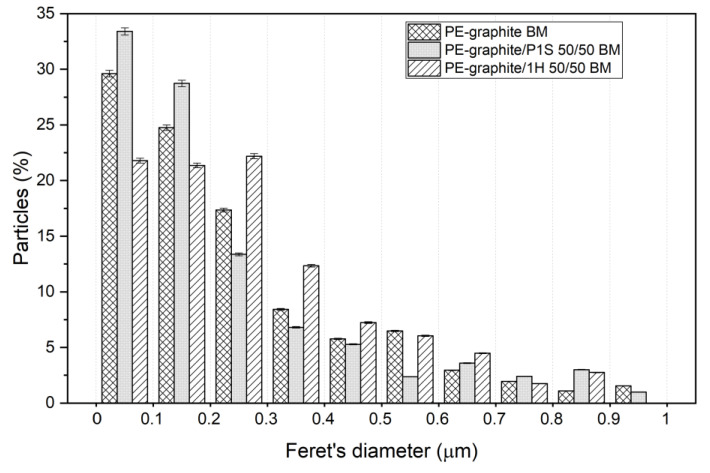
Particle size distribution of PE-graphite, PE-graphite/P1S 50/50 and PE-graphite/1H 50 BM. Each histogram refers to a Feret’s diameter included in a range of 0.1 micron.

**Figure 6 polymers-15-02770-f006:**
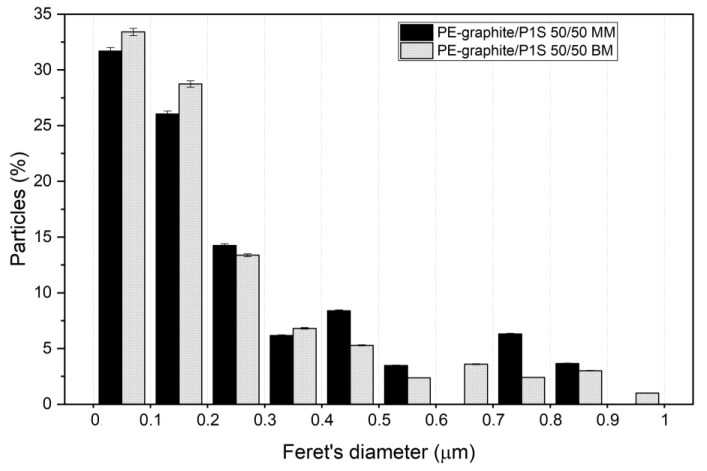
Particle size distribution of PE-graphite/P1S 50/50 MM and PE-graphite/P1S 50/50 BM. Each histogram refers to a Feret’s diameter included in a range of 0.1 micron.

**Table 1 polymers-15-02770-t001:** Inflection point temperatures and final degradations.

Samples	Temperature (°C)	Weight Loss (%)
Graphite	not detected	2.1
Graphite/P1S 90/10 BM	291.6	8.4
Graphite/P1S 90/10 MM	283.8	10.0
Graphite/P1S 50/50 BM	335.6	41.7
Graphite/1H 50/50 BM	249.3	44.7
Graphite/1H 50/50 MM	253.4	46.3
Graphite/P1S 50/50 MM	336.1	49.8
P1S	355.7	98.1
1H	241.9	97.3

## Data Availability

The data presented in this study are available in the Appendix A.

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
