# Peer review of "Mechano Chemical Compatibilization of Polyethylene with Graphite by Means of a Suitable Ester"

_polymers, 2023, doi:10.3390/polym15132770_

Round 1

Reviewer 1 Report

-Page 4, Line 146-153 Why was only two ratios selected for graphite/PIS (90-10 and 50-50). Why not the third ratio say 10-90?

- The explanation on the TGA Profile needs to be deepened. The various temperature changes for the samples can be summarized in a Table.

-Fig 4, the scale bar for the magnification is not visible

-Overall, the discussions on the presented results need to be deepened., with more recent and relevant citations.

Overall the English quality is fair and can be improved.

Author Response

Comments of Reviewer 1

  1. Page 4, Line 146-153 Why was only two ratios selected for graphite/PIS (90-10 and 50-50). Why not the third ratio say 10-90?
  2. The explanation on the TGA Profile needs to be deepened. The various temperature changes for the samples can be summarized in a Table.
  3. Fig 4, the scale bar for the magnification is not visible.
  4. Overall, the discussions on the presented results need to be deepened with more recent and relevant citations.

Answers to Reviewer 1 

The authors express their gratitude to the reviewer for her/his positive comments.

  1. The ratio graphite/P1S 10-90 has not been selected because the aim of our work was to obtain the compatibilization of the polyethylene matrix with graphite. In this perspective, we don’t want the amount of P1S ester to exceed the amount of filler.
  2. The authors thank the reviewer for the interesting comment. We added a table in the manuscript containing the temperatures corresponding to the maximum degradation rate assessed at the inflection point and weight loss (%) referred to the final degradation.

Table 1. Inflection point temperatures and final degradations.

Samples

Temperature (°C)a

Weight loss (%)b

Graphite

n.d.c

2.1

Graphite/P1S 90/10 BM

291.6

8.4

Graphite/P1S 90/10 MM

283.8

10.0

Graphite/P1S 50/50 BM

335.6

41.7

Graphite/1H 50/50 BM

249.3

44.7

Graphite/1H 50/50 MM

253.4

46.3

Graphite/P1S 50/50 MM

336.1

49.8

P1S

355.7

98.1

1H

241.9

97.3

aThe temperatures are referred to the maximum degradation rate assessed at the inflection point.

bWeight loss (%) referred to the final degradation.

cnot detected.

  1. The figure 4 has been modified as requested.
  1. We added more recent and relevant references:
  2. Geyer, R.; Jambeck, J.R.; Law, K.L. Production, Use, and Fate of All Plastics Ever Made. Adv. 2017, 3, e1700782, doi:10.1126/sciadv.1700782
  3. Ronca, S. Polyethylene. In Brydson’s Plastics Materials; Elsevier, 2017; pp. 247–278 ISBN 978-0-323-35824-8.
  4. Patel, R.M. Multilayer Flexible Packaging; Elsevier, 2016; ISBN 978-0-323-37100-1.
  5. Jubinville, D.; Esmizadeh, E.; Saikrishnan, S.; Tzoganakis, C.; Mekonnen, T. A Comprehensive Review of Global Production and Recycling Methods of Polyolefin (PO) Based Products and Their Post-Recycling Applications. Sustainable Materials and Technologies 2020, 25, e00188, doi:10.1016/j.susmat.2020.e00188.
  6. López-González, M.; Flores, A.; Marra, F.; Ellis, G.; Gómez-Fatou, M.; J. Salavagione, H. Graphene and Polyethylene: A Strong Combination Towards Multifunctional Nanocomposites. Polymers 2020, 12, 2094, doi:10.3390/polym12092094.

Reviewer 2 Report

In this submitted manuscript (polymers-2449687), Drs. Maria Rosaria Acocella, Annaluisa Mariconda, and co-authors studied the mechano-chemically compatibilization of graphite in polyethylene matrix through the pyren-1-yl-stearate (P1S). A series of P1S/graphite fillers have been prepared with different mass ratios and mixing methods (manual or ball milling). After being added to polyethylene, their stability, interaction, and morphology were studied through TGA, RX, and SEM. It was found that the P1S could improve the dispersion of carbon filler in polymer via the non-covalent compatibilization, specifically, the π-π stacking interaction within the aromatic rings and Van der Waals interactions in the long aliphatic stearyl chain. It proves that ball milling is a more effective way than manual mixing to yield the filler which is better compatible with polyethylene.

It is important to develop an efficient compatibilization method for graphite and low-density polyethylene, and the findings reported in this manuscript would appeal to the broad readership of Polymers, especially those who focus on studying modifying properties of polyethylene. However, some necessary descriptions and discussions were missed, and the language of this manuscript should be thoroughly improved. Based on those, major revisions are needed before further consideration.

1. The carbon fillers prepared in this manuscript are P1S/graphite with a mass ratio of 90/10 and 50/50. With such a high ratio of P1S, would it lead to submerging the unique properties of graphite after adding it to polyethylene?

2. In the preparation of P1S on lines 115-116, the product was obtained after evaporating the filtrate under the vacuum. Is there any further purification step here? The reviewer doubts if a simple filtration step could give the pure product for direct use without any other purifications. Also, in the same paragraph, on line 113, is the reaction time “3.5 hours”? It is not correct to write “3,5 hours”.

3. On lines 174-175, it was assumed that the π-π stacking interaction was formed between the pyrene group and the aromatic rings of graphite. Is there any evidence or characterization that can prove this? That will make this conclusion more convictive.

4. On line 51, the correct name of DMF should be N, N-dimethylformamide. The letter N in the name of NMP should be italic – N-methyl pyrrolidine.

5. The conclusion section is a little too tedious. Please rewrite the conclusion section to make it more concise and highlight the key findings reported in this manuscript.

6. The language needs thorough improvement. There are many sentences with grammatical errors or hard to understand. For example, on line 61, instead of writing “…interactions is…”, it should be “…interactions are…”. On lines 227-228, that sentence missed its subject. It makes no sense to say “The thermogravimetric curves of samples containing 90 wt% of graphite and 10 wt% of the P1S ester up to over 500 °C” on lines 236-237. What is “up to over 500oC”? The subject of this sentence is missed again. Other sentences which are hard to understand include the ones on lines 247, 250, 285, 325, etc.

7. For the sentence on lines 371-372, it looks like to be the instruction of a template for authors. No need to keep it in the manuscript. The same comment is for the paragraph of the “Funding” section.

The language of this manuscript needs to be thoroughly improved. Many sentences with grammatical errors were found.

Author Response

Comments of Reviewer 2

In this submitted manuscript (polymers-2449687), Drs. Maria Rosaria Acocella, Annaluisa Mariconda, and co-authors studied the mechano-chemically compatibilization of graphite in polyethylene matrix through the pyren-1-yl-stearate (P1S). A series of P1S/graphite fillers have been prepared with different mass ratios and mixing methods (manual or ball milling). After being added to polyethylene, their stability, interaction, and morphology were studied through TGA, RX, and SEM. It was found that the P1S could improve the dispersion of carbon filler in polymer via the non-covalent compatibilization, specifically, the π-π stacking interaction within the aromatic rings and Van der Waals interactions in the long aliphatic stearyl chain. It proves that ball milling is a more effective way than manual mixing to yield the filler which is better compatible with polyethylene.

It is important to develop an efficient compatibilization method for graphite and low-density polyethylene, and the findings reported in this manuscript would appeal to the broad readership of Polymers, especially those who focus on studying modifying properties of polyethylene. However, some necessary descriptions and discussions were missed, and the language of this manuscript should be thoroughly improved. Based on those, major revisions are needed before further consideration.

  1. The carbon fillers prepared in this manuscript are P1S/graphite with a mass ratio of 90/10 and 50/50. With such a high ratio of P1S, would it lead to submerging the unique properties of graphite after adding it to polyethylene?
  2. In the preparation of P1S on lines 115-116, the product was obtained after evaporating the filtrate under the vacuum. Is there any further purification step here? The reviewer doubts if a simple filtration step could give the pure product for direct use without any other purifications. Also, in the same paragraph, on line 113, is the reaction time “3.5 hours”? It is not correct to write “3,5 hours”.
  3. On lines 174-175, it was assumed that the π-π stacking interaction was formed between the pyrene group and the aromatic rings of graphite. Is there any evidence or characterization that can prove this? That will make this conclusion more convictive.
  4. On line 51, the correct name of DMF should be N, N-dimethylformamide. The letter N in the name of NMP should be italic – N-methyl pyrrolidine.
  5. The conclusion section is a little too tedious. Please rewrite the conclusion section to make it more concise and highlight the key findings reported in this manuscript.
  6. The language needs thorough improvement. There are many sentences with grammatical errors or hard to understand. For example, on line 61, instead of writing “…interactions is…”, it should be “…interactions are…”. On lines 227-228, that sentence missed its subject. It makes no sense to say “The thermogravimetric curves of samples containing 90 wt% of graphite and 10 wt% of the P1S ester up to over 500 °C” on lines 236-237. What is “up to over 500oC”? The subject of this sentence is missed again. Other sentences which are hard to understand include the ones on lines 247, 250, 285, 325, etc.
  7. For the sentence on lines 371-372, it looks like to be the instruction of a template for authors. No need to keep it in the manuscript. The same comment is for the paragraph of the “Funding” section.

Answers to Reviewer 2

The authors express their gratitude to the reviewer for her/his positive comments.

  1. Thanks for your comment. As it is well known, good filler dispersion in a such polymer matrix can ensure the improvement of the physical or mechanical properties of the obtained composite. Graphite typically exhibits a strong tendency to agglomerate, which prevents it from dispersing uniformly. The goal of this work is to use a compatibilizer that promotes graphite exfoliation and interacts with the polymer chain to provide the right dispersion.

As clearly shown in the SEM images and the particle size distribution reported in figure 5, the Graphite/P1S 50/50 ball milled can improve significantly the filler dispersion in polyethylene. As a result, we do not expect P1S to submerge graphite's unique properties, but rather to observe them improving. Of course, further studies are in progress to better understand the influence of the P1S on the graphite properties and will be reported in due course.

  1. As you suggested, 3,5 hours on line 113 has been replaced with 3.5 hours. About the preparation of P1S, that is described on lines 115-116, we report the work-up here in a more explicit way.

After stripping the reaction solvent, pentane was added to the reaction crude in order to remove the pyridinium chloride: the ester was soluble in pentane while the pyridinium salt was not. So the filtration has been carried out and the salt was collected on the filter paper. P1S ester was recovered from the pentane solution and obtained in the form of a pale yellow solid after the removal of the solvent under vacuum.

  1. The π-π stacking interaction between the pyrene group and the aromatic rings of graphite can be supported by TGA and X-Ray diffraction analyses.

As shown on page 6, the TGA profiles for graphite/P1S 50/50 manually grinding and ball milling, show different weight loss, 50wt% and 40wt%, respectively. The ball milling allows 10wt% of P1S to be intercalated, protecting it during thermal degradation. The intercalation can possibly occur because of the π-π stacking interaction. Furthermore, the same behavior has been observed with Graphite/1H (hydroxypyrene) 50/50 manual and ball milling mixed.

This aspect is described on page 6, lines 240-258.

Comparison of the X-ray diffraction patterns of graphite sample and graphite/P1S 50/50 ball milled sample provides additional information. Specifically, the milling action on graphite causes a slight increase in amorphous halo, reduction of half-height width and shift of 002 reflection to higher theta, due to the possible aggregation of graphene layers. Conversely, Graphite/P1S 50/50 ball milled shows an increase in amorphous halo and a broadening of 002 reflection due to the exfoliation and intercalation of ester in the graphitic structure.

This aspect is described on page 7 lines 276-288 and page 8 lines 289-296.

  1. Dimethylformamide has been replaced with N, N-dimethylformamide. The letter N in the name of N-methyl pyrrolidine has been changed in N-methyl pyrrolidine.
  2. The authors thank the reviewers for the right suggestion. In the manuscript we reported the conclusions in a more concise version.
  3. All changes have been made, as you suggested.

On line 61 “interactions is” has been replaced with “interactions are”.

On lines 227-228 the sentence “moreover, for a convenient comparison, are also reported the thermograms of graphite, 1H, and pyren-1-yl-stearate.” has been changed in “moreover, for a convenient comparison, the thermograms of graphite, 1H, and pyren-1-yl-stearate are also reported’’

On lines 236-237, “The thermogravimetric curves of samples containing 90 wt% of graphite and 10 wt% of the P1S ester up to over 500 °C” has been replaced with “The thermogravimetric curves of samples containing 90 wt% of graphite and 10 wt% of the P1S ester” (up to over 500% has been removed”).

On line 247 the sentence “even in this case, the trend is respected because the curve of the milled sample is just above, highlighting a modest difference respect to the unmilled one” has been modified in “even in this case, the trend is respected because the curve of the milled sample is above that of the manual mixed one”

On line 250 the sentence: “From the thermogravimetric analyses it is possible to deduce that the P1S ester probably intercalates the graphene layers resulting strongly bonded only by applying the ball milling technique” has been replaced with “From the thermogravimetric analyses it is possible to deduce that the P1S ester is capable of intercalate the graphene layers only if the ball milling technique is used.”

On line sentence 285 “Here we show, on the other hand, that the in presence of a high surface area graphite (currently used in our studies) milling results in a slight increase in amorphous halo and simultaneous possible aggregation of graphene layers, with reduction of half-height width and a shift of 002 reflection to higher theta” has been replaced with “Our results suggest, on the other hand, that milling graphite with a high surface area (currently used in our studies) increases the amorphous halo and simultaneously enables the aggregation of graphene layers, thereby reducing half-height width and shifting 002 reflection to higher theta”

On line 325 the sentence “As a result, the particle size distribution of PE-graphite, PE-graphite/P1S 50/50 and PE-graphite/1H 50/50 BM (Figure 5) clearly shows the better dispersion of the PE-graphite/P1S 50/50 BM, being higher the percentage of particles size in the range 0-0.2 micron” has been changed in “As a result, the particle size distribution of PE-graphite, PE-graphite/P1S 50/50 and PE-graphite/1H 50/50 BM (Figure 5) clearly shows the better dispersion of the PE-graphite/P1S 50/50 BM, since the percentage of particles size in the range 0-0.2 micron is higher.

  1. The template instructions have been deleted in all the paragraphs.

Reviewer 3 Report

The manuscript entitled “Mechano chemical compatibilization of polyethylene with graphite by means of a suitable ester” describes a new method for the compatibilization of polyethylene with graphite, where pyren-1-yl-stearate (P1S) was synthesized and mixed with graphite to achieve better dispersibility. It was found that PS1 enhanced the dispersibility of the graphite in the polyethylene matrix via noncovalent interactions. In particular, pi-pi interactions took place between the pyrenyl groups of PS1 and graphite, while the stearyl chain participated in van der Waals interactions. The authors prepared PS1/graphite fillers at weight ratios of 90/10 and 50/50 via manual and ball mixing approaches, respectively. It was found that ball milling was more effective in promoting pi-pi interactions than manual mixing. XPS characterization revealed that both exfoliation and intercalation occurred when ball milling was used to mix the P1S ester with graphite.

            The research described in this manuscript is fascinating and fits well within the scope of Polymers. For the most part, the manuscript is well-written, and the conclusions are generally well-supported by the characterization data. Overall, I believe that this submission is suitable for publication pending minor revisions, such as those outlined below.

Line 18-19: The phrase “and stearyl chain provides Van der Waals interaction with polymer chain” seems to be a little vague.. Also “Van der Waals” should be changed to “van der Waals”.

Lines 33-36: reference citations may be needed for the first couple of sentence in the introduction describing the advantages of polyolefins.

Line 83: “All reagents were bought by Merck or TCI Chemicals” should be changed to “All reagents were bought from Merck or TCI Chemicals” or “All reagents were purchased from Merck or TCI Chemicals”.

Line 113: “3,5 hours” should possibly be changed to “3.5 hours”.

Line 136: There seems to be an empty set of parenthesis in the phrase “ImaeJ [18][].”. It may also be necessary to check the name of the image analysis software used here. Should this be “ImageJ” instead?

Line 203: “this may prevents” can be changed to “this may prevent”.

Line 217: “results of ball milling technique can” can possibly be changed to ”results of the ball milling technique can” or “results obtained via the ball milling technique can”.

Line 256: The phrase “molecules of hydroxy-polycyclic aromatic are” may be unclear.

Line 314: “as characteristic size” can possibly be changed to “as a characteristic size” or “as the characteristic size”.

Line 346: “Van der Waals interactions” can be changed to “van der Waals interactions”.

Overall, the manuscript is well-written and clear. There are some areas that could benefit from minor polishing, which I have outlined in further detail in the comments to the authors. 

Author Response

Comments of Reviewer 3

The manuscript entitled “Mechano chemical compatibilization of polyethylene with graphite by means of a suitable ester” describes a new method for the compatibilization of polyethylene with graphite, where pyren-1-yl-stearate (P1S) was synthesized and mixed with graphite to achieve better dispersibility. It was found that PS1 enhanced the dispersibility of the graphite in the polyethylene matrix via noncovalent interactions. In particular, pi-pi interactions took place between the pyrenyl groups of PS1 and graphite, while the stearyl chain participated in van der Waals interactions. The authors prepared PS1/graphite fillers at weight ratios of 90/10 and 50/50 via manual and ball mixing approaches, respectively. It was found that ball milling was more effective in promoting pi-pi interactions than manual mixing. XPS characterization revealed that both exfoliation and intercalation occurred when ball milling was used to mix the P1S ester with graphite.

The research described in this manuscript is fascinating and fits well within the scope of Polymers. For the most part, the manuscript is well-written, and the conclusions are generally well-supported by the characterization data. Overall, I believe that this submission is suitable for publication pending minor revisions, such as those outlined below.

  1. Line 18-19: The phrase “and stearyl chain provides Van der Waals interaction with polymer chain” seems to be a little vague.. Also “Van der Waals” should be changed to “van der Waals”.
  2. Lines 33-36: reference citations may be needed for the first couple of sentence in the introduction describing the advantages of polyolefins.
  3. Line 83: “All reagents were bought by Merck or TCI Chemicals” should be changed to “All reagents were bought from Merck or TCI Chemicals” or “All reagents were purchased from Merck or TCI Chemicals”.
  4. Line 113: “3,5 hours” should possibly be changed to “3.5 hours”.
  5. Line 136: There seems to be an empty set of parenthesis in the phrase “ImaeJ [18][].”. It may also be necessary to check the name of the image analysis software used here. Should this be “ImageJ” instead?
  6. Line 203: “this may prevents” can be changed to “this may prevent”.
  7. Line 217: “results of ball milling technique can” can possibly be changed to ”results of the ball milling technique can” or “results obtained via the ball milling technique can”.
  8. Line 256: The phrase “molecules of hydroxy-polycyclic aromatic are” may be unclear.
  9. Line 314: “as characteristic size” can possibly be changed to “as a characteristic size” or “as the characteristic size”.
  10. Line 346: “Van der Waals interactions” can be changed to “van der Waals interactions”.

Answers to Reviewer 3

The authors express their gratitude to the reviewer for her/his positive comments.

  1. The phrase “and stearyl chain provides Van der Waals interaction with polymer chain” on lines 18-19 has been modified in “and stearyl chain provides van der Waals interaction with polymer chain (specifically London dispersion forces).”
  2. We added more recent and relevant references: [1] Geyer, R.; Jambeck, J.R.; Law, K.L. Production, Use, and Fate of All Plastics Ever Made. Adv. 2017, 3, e1700782, doi:10.1126/sciadv.1700782. [2] Ronca, S. Polyethylene. In Brydson’s Plastics Materials; Elsevier, 2017; pp. 247–278 ISBN 978-0-323-35824-8. [3] Patel, R.M. Multilayer Flexible Packaging; Elsevier, 2016; ISBN 978-0-323-37100-1. [4] Jubinville, D.; Esmizadeh, E.; Saikrishnan, S.; Tzoganakis, C.; Mekonnen, T. A Comprehensive Review of Global Production and Recycling Methods of Polyolefin (PO) Based Products and Their Post-Recycling Applications. Sustainable Materials and Technologies 2020, 25, e00188, doi:10.1016/j.susmat.2020.e00188.  [5] López-González, M.; Flores, A.; Marra, F.; Ellis, G.; Gómez-Fatou, M.; J. Salavagione, H. Graphene and Polyethylene: A Strong Combination Towards Multifunctional Nanocomposites. Polymers 2020, 12, 2094, doi:10.3390/polym12092094.
  3. On line 83 the sentence “All reagents were bought by Merck or TCI Chemicals” has been changed in “All reagents were bought from Merck or TCI Chemicals”.
  4. “3,5 hours” on line 113 has been replaced with “3.5 hours”.
  5. On line 136 “ImaeJ” has been replaced with “ImageJ” and the set of parenthesis has been deleted.
  6. On line 203 “prevents” has been modified in “prevent”.
  7. On line 217 the sentence “results of ball milling technique can” has been replaced with “results of the ball milling technique can”.
  8. Line 256: the sentence has been modified as ‘In both samples is evident the almost complete degradation of the hydroxypyrene at temperatures around 270 °C.’
  9. On line 314 the sentence “as characteristic size” has been changed in “as a characteristic size”.
  10. On line 346 the letter V in “Van der Waals interactions” has been rewritten with a lower case letter v.

Round 2

Reviewer 2 Report

In the revised manuscript, the authors discussed and provided detailed explanations to the comments from the reviewers. The quality of the manuscript has been improved and I'm happy to recommend accepting it in its current version.